# Optimising Building Energy and Comfort Predictions with Intelligent Computational Model

Salah Alghamdi [1], Waiching Tang [2,*], Sittimont Kanjanabootra [2] and Dariusz Alterman [3]

1   Department of Building Engineering, College of Architecture and Planning, Imam Abdulrahman Bin Faisal University, Dammam 31451, Saudi Arabia; saghamdi@iau.edu.sa
2   School of Architecture and Built Environment, The University of Newcastle, University Drive, Callaghan, NSW 2308, Australia; sittimont.kanjanabootra@newcastle.edu.au
3   School of Science, Technology and Engineering, The University of the Sunshine Coast, 90 Sippy Downs Drive, Sippy Downs, QLD 4556, Australia; dalterman@usc.edu.au
*   Correspondence: patrick.tang@newcastle.edu.au; Tel.: +61-(2)4-921-7246

**Abstract:** Building performance prediction is a significant area of research, due to its potential to enhance the efficiency of building energy management systems. Its importance is particularly evident when such predictions are validated against field data. This paper presents an intelligent computational model combining Monte Carlo analysis, Energy Plus, and an artificial neural network (ANN) to refine energy consumption and thermal comfort predictions. This model addresses various combinations of architectural building design parameters and their distributions, effectively managing the complex non-linear relationships between the response variables and predictors. The model's strength is demonstrated through its alignment with $R^2$ values exceeding 0.97 for both thermal discomfort hours and energy consumption during the training and testing phases. Validation with field investigation data further confirms its accuracy, demonstrating average relative errors below 2.0% for total energy consumption and below 1.0% for average thermal discomfort hours. In particular, an average underestimation of $-12.5\%$ in performance discrepancies is observed when comparing the building energy simulation model with field data, while the intelligent computational model presented a smaller overestimation error (of $+8.65\%$) when validated against the field data. This discrepancy highlights the model's potential and reliability for the simulation of real-world building performance metrics, marking it as a valuable tool for practitioners and researchers in the field of building sustainability.

**Keywords:** artificial neural networks; architectural building design parameters; energy consumption; educational buildings; thermal comfort





## 1. Introduction

Buildings must be resilient to climate change, as well as adaptable, flexible, and durable with respect to increased occupancy levels, thus increasing thermal comfort and decreasing the energy consumption of buildings [1]. Global energy consumption increased by an estimated 56% between 2014 and 2020, resulting in significant negative impacts on the environment [2–4]. The buildings sector accounts for around 50.0% of global energy consumption, and this percentage is expected to increase to 60.0% in the near future [5,6]. Numerous studies have shown that the air conditioning system is the primary contributor to energy consumption in buildings [2,7–9], consuming up to 38.0% of the overall energy consumption in the building sector [9,10]. This percentage is rising due to climate change and individuals are spending more time inside [3,9]. Implementing effective energy consumption methods in sectors with high demand, such as the buildings sector, could assist in reducing this issue [4]. The energy consumption method presented in this article integrates architectural design parameters with evaluation and decision-making processes during the initial design stage. The methodology is based on the segmentation of

parameter performance into scenarios, each associated with its specific energy consumption based on their possible operating states. Energy can be simultaneously saved and utilised more efficiently while enhancing the thermal comfort level [7]. Through precise prediction of energy usage and thermal comfort levels at the conceptual design phase, a quantitative basis for the creation of energy-efficient solutions can be established. Thermal comfort prediction models offer an opportunity to respond to both individuals' comfort needs and energy efficiency [11]. Machine learning has great potential for improving prediction models for the thermal comfort and performance of building heating, ventilation, and air conditioning (HVAC) systems, aiding designers in developing energy savings, and indoor environmental quality [12–15].

Artificial neural networks (ANNs) find extensive application across several domains, including prediction and inverse modelling of complex systems with various inputs and outputs [16]. The ANN excels as a machine learning technique for discerning the relationships between designated input and output variables [17], achieved through mimicking the way in which the human brain assimilates information from the environment through a learning process [18]. When there are several input variables, the most efficient way to predict two objectives is through the use of an ANN model [16,19]. ANNs have been shown to outperform multiple regression for data analysis in several problem domains [20]. Additionally, the abilities of ANNs extend beyond those of regression techniques, such as having the capability to deal with non-linear relationships and missing data [21]. The network is configured as follows [22]: (a) the network is provided with inputs; (b) computed output values are compared with expected outputs, necessitating a database of output values corresponding to input parameter vectors for network training; and (c) the network's biases and weights are adjusted to minimize the disparity between expected and estimated output values [22–24].

Of the training methods that can be used to optimise the weight and bias values, back-propagation is one of the most-used [25,26]. ANNs are often used to improve building performance by modelling systems that are challenging to model with energy software [16,27]. Yildiz et al. (2017) have assessed the effectiveness of an ANN with Bayesian regulatory back-propagation in estimating the University of New South Wales (UNSW) building and campus power consumption using actual building data. Another study has described an approach for comparing simulated findings to actual measurements, using one week of recorded data to generate an entire year's worth of heating and cooling electricity use using an ANN [28]. The projected results demonstrated a good fit with the mathematical model, with an absolute mean error of 0.9%. An ANN model has been designed to regulate the set-point temperature in an effort to improve thermal comfort and reduce energy, resulting in substantial 25.0% energy savings [29,30]. Another study has developed an ANN model for assessing the energy efficiency of public buildings. The ANN was trained utilising a database generated through the simulation of multiple building models under varying climatic conditions [31]. In [32], the total energy consumption was predicted using nine input variables associated with external climate data, four of which were related to the building envelope and one to the day type (weekday or weekend); they utilised an ANN for this purpose. In [33], a comparative analysis of ANN and random forest models was conducted to determine the electricity consumption associated with HVAC systems in hotels. The outcomes demonstrated that the ANN outperforms the random forest by a small margin. A significant amount of research on building energy has concentrated on how to predict energy use and/or thermal comfort to enhance building performance [34–36]; however, the prediction of energy savings and thermal comfort assessments in air-conditioned educational buildings in Australia through multiple architectural building design parameters remains limited at present.

In general, energy-saving strategies can be classified into two categories: passive and active solutions [37]. Active solutions entail enhancing HVAC systems, lighting, and other building services systems. On the other hand, passive solutions focus on using energy-efficient architectural parameters like building envelopes and roofs to reduce reliance on

active solutions [8]. Therefore, the following research question was formulated: How can energy consumption and thermal comfort levels be predicted through multiple architectural building design parameters (ABDPs) to improve building performance? In this study, an ANN model is used to predict the thermal discomfort hours (TDHs) of students and entire building energy consumption (EC), as output values, under any given combination of ABDPs, including both active and passive parameters (15 input values), through using the neural network toolbox in MatLab R2021a software. The considered ABDPs are the glazing type, roof window openings ratio (skylight), window-to-wall ratio, local shading type, mechanical ventilation rate per area, infiltration rate, crack template (airtightness), thermal mass level, external wall construction, roof construction, building orientation, building location, cooling set-point temperature, heating set-point temperature, and occupancy density. The selection of these 15 parameters was based on the fact that designers have the potential to effectively regulate them while remaining in the conceptual design stage of development. For example, adjusting the load shading type parameter during the conceptual design stage rather than post-construction is a more economical method for improving occupant thermal comfort and reducing EC [38]. Delaying efforts to enhance a building's performance beyond the conceptual design stage signifies a wasted opportunity to employ a more economically efficient alternative [39]. Through inputting these ABDPs, this ANN model can assist architects in estimating building performance at an early stage of architectural design, thus reducing the non-essential modelling time and enhancing the sustainability performance of architectural design.

## 2. Materials and Methods

In this section, the proposed procedure undertaken to develop the ANN model to predict EC and TDHs for any given combination of the identified ABDPs in the conceptual design stage is detailed. The present study proposes a workflow integrating a building energy simulation (BES) model to generate an energy consumption (EC) and TDHs database. The DesignBuilder (DB) software was used to simulate scenarios characterised by different ABDPs. Utilising comprehensive energy simulation, the efficiency of educational buildings in hot summer and cold winter environments under the various ABDPs was evaluated. In addition, Monte Carlo analysis (MCA) was adopted to cover all probability scenarios under the input ABDPs. For each input parameter, 2000 samples were generated using the Latin hypercube sampling (LHS) technique, based on the chosen distribution. To further validate the BES and ANN models in terms of improving energy savings and the thermal comfort of students, the field investigation method was applied through conducting subjective and objective measurements. Figure 1 illustrates the methodology employed in the study.

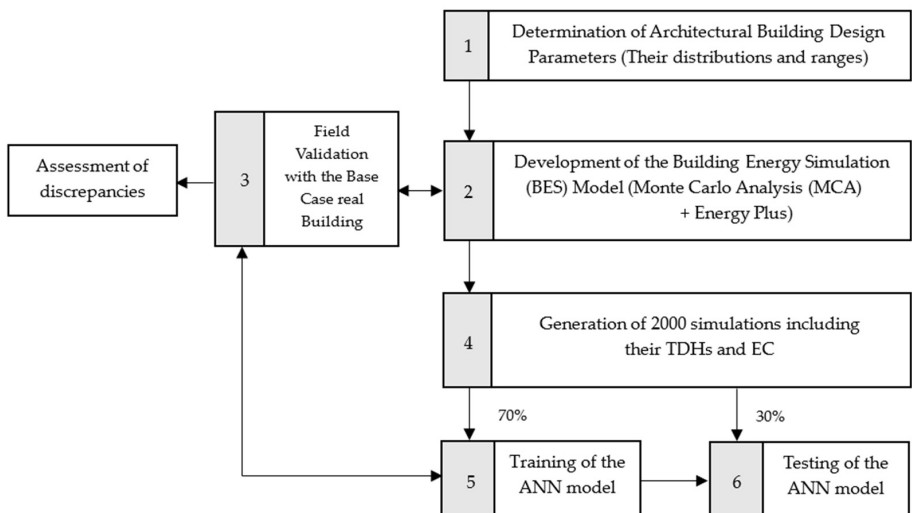

**Figure 1.** Flow chart depicting research process.

### 2.1. Determination of Architectural Building Design Parameters and Monte Carlo Analysis

Heat gain or loss through active and passive strategies involving ABDPs (e.g., cooling and heating set-point temperature, walls, windows, and mechanical ventilation rate per area) are critical factors in regulating the indoor thermal environment of buildings [40,41]. The EC required for heating and cooling, in order to maintain appropriate levels of thermal comfort, is consequently influenced by the indoor thermal environment. The 15 most significant ABDPs affecting the thermal comfort and EC of buildings were determined, as shown in Table 1, following a comprehensive review of the relevant literature [39,42]. Additionally, the selection of these parameters was based on the understanding that they have been investigated individually across various applications, involving diverse climatic zones, types of buildings, building occupancy, and outputs including thermal comfort and/or EC.

**Table 1.** Distribution types of selected ABDPs.

| | Input Parameter | Distribution Type | Input Units | Parameter Values |
|---|---|---|---|---|
| 1 | Glazing type | Discrete | (U-value) $W/m^2K$ | [Single bronze (3 mm) (6.25), single Ref-C-H clear (6 mm) (5.30), double LoE (e3 = 0.1) Clr (3 mm/13 mm) air (2.70), double bronze (3 mm/6 mm) air (3.20), double clear (6 mm/13 mm) air (1.80)] |
| 2 | Roof window openings ratio (skylight) | Normal | % | [3–17] |
| 3 | Window-to-wall ratio | Normal | % | [5–75] |
| 4 | Local shading type | Discrete | m | [Louvres with projections of 0.5 m, Louvres with projection of 1.0 m, Louvres with projection of 1.5 m, overhangs with projection of 1.0 m, overhangs with projection of 1.5 m, overhangs with projection of 2.0 m, no shading] |
| 5 | Mechanical ventilation rate per area | Normal | $l/s/m^2$ | [2–8] |
| 6 | Infiltration rate | Normal | ACH [a] | [0.4–2] |
| 7 | Crack template (airtightness) | Discrete | - | [Excellent, Good, Medium, Poor, Very Poor] |
| 8 | Thermal mass level | Discrete | - | [Low, Medium, High] |
| 9 | External wall construction | Discrete | (U-value) $W/m^2K$ | [Brickwork single-leaf construction light plaster (1.95), brick air lightweight concrete block and lightweight plaster (0.95), brick mineral insulation Thermolite block and lightweight plaster (0.40), brick/block wall (insulated to 1985 regs) (0.35), uninsulated lightweight wall (metal-clad) (2.75)] |
| 10 | Roof construction | Discrete | (U-value) $W/m^2K$ | [25 mm stone chippings on 19 mm asphalt on 40 mm screed (3.45), 19 mm asphalt on 13 mm fibreboard (2.60), 19 mm asphalt on 13 mm screed on 50 mm wood wool slab (1.45), 6 mm lightweight metallic cladding (6.22), 75 mm of concrete deck (0.25), green roof construction (0.23)] |
| 11 | Building orientation | Normal | Angle (°θ) | [0–315] |
| 12 | Building location | Discrete | - | [Newcastle, Sydney] |
| 13 | Cooling set-point temperature | Normal | °C | [19–28] |
| 14 | Heating set-point temperature | Normal | °C | [17–23] |
| 15 | Occupancy density | Normal | people/$m^2$ | [0.1–0.5] |

Note: [a] air change per hour.

The estimation of limits on the variance of ABDPs was conducted to ascertain the optimal parameter values and probability density functions [43]. The range values (or distribution characteristics) of the ABDPs were derived from the possible values in architectural practice revealed through the literature review. Table 1 shows the two distribution characteristics employed in building design: discrete and normal [44,45]. In discrete distribution, data (e.g., the thermal mass level) can only take on certain values [46,47]. Normal distribution allows data (e.g., the set-point temperature) to take any value within a given range (which may be infinite) [46,47]. In the present study, the included distributions associated with each ABDP were determined according to previous investigations or practical applications. The interval width is used to indicate a wide variety of intervals for the distribution, while also increasing the model's accuracy throughout the evaluation stage [48].

Utilising the principles of inferential statistics, Monte Carlo analysis (MCA) is a technique employed to approximate the value of an unknown quantity [49]. MCA-based simulation is a common technique utilised to assess the energy efficiency of buildings [50,51], as, compared to other approaches (e.g., full factorial numerical integration, stochastic Galerkin, and discrete projection), this method is very intuitive and straightforward to apply [52]. The sampling-based method is widely acknowledged as the most reliable uncertainty technique [53]. Moreover, the MCA can be implemented in the majority of simulation environments and can handle a variety of probability functions for the input variables.

MCA was adopted to cover all probability scenarios for the input ABDPs. To ensure sufficient accuracy in the uncertainty analyses (UAs), 2000 samples were carried out for each input parameter depending on the specified distribution, using the Latin hypercube sampling (LHS) technique. This study utilized UA to support the design process by quantifying the impact of various parameters on design objectives (the log normal or normal distribution and range). This method ensures that all possible cases are covered when developing the ANN model.

### 2.2. Development of the Building Energy Simulation Model and Validation through Field Investigation Data Collection

The effective modelling and visualisation of the complexity of an indoor thermal environment is typically achieved through the application of simulation software [54,55]. For the purpose of evaluating the EC and thermal comfort of various design alternatives for proposed or existing buildings, use of the (DB) software is now common practice [56]. DB is a readily accessible software developed for the purpose of simulating and analysing energy flows through a dynamic building energy system, including building modelling, heating, cooling, ventilation, and other related processes [57]. The software was developed in the 1990s through combining the BLAST and DOE-2 simulation engines [58]. This tool was designed to provide the capability to enter variable geometries, access a wide range of material libraries, and utilise load profiles [59,60].

The meteorological data for all calculations was obtained from an EnergyPlus meteorological (EPW) file that includes the overall climatic characteristics of Newcastle and Sydney [61–63]. The software utilises the definitions provided by the American Society of Heating, Refrigerating, and Air-Conditioning Engineers (ASHRAE) for its computations, and employs several approaches for data management [64]. The heat balance approach was selected as the calculation method, due to its more basic nature compared to the weighting factor approach [58,65]. This study used the DB software (version 6.0.1.019) with a user-friendly interface to simulate various ABDPs and analyse the overall energy performance estimates for the entire building.

### 2.2.1. Case Study Design and Climate

For the BES model adopted in this study, the process of simulation utilised a single-storey building at The University of Newcastle, Australia, as the reference building and base case model. The building is classified as a generic representation of a building in

climate zone 5, according to the Australian Building Codes Board. It is located in a warm climate and possesses architectural characteristics commonly encountered in educational buildings. The building materials and construction methods used for walls, floors, and windows are typical for this type of building. The architectural structure has a square configuration, measuring 10.5 m in length, 10.5 m in width, and 3.5 m in height. Its overall volume is estimated to be around 386 m$^3$, as seen in Figures 2 and 3. DesignBuilder (DB) was used to simulate scenarios characterised by different ABDPs. Comprehensive energy simulation was utilized to assess the performance of educational spaces under various ABDPs in both hot summer and cold winter environments. This was completed to attain an optimal balance between thermal comfort and energy consumption. The environmental factors for the zone were carefully chosen to establish a realistic energy simulation, allowing for adjustable indoor environmental conditions that could be altered throughout the year-long simulation process. The weather data for all computations were obtained from an EnergyPlus weather (EPW) file that specifically captured the climatic characteristics of Newcastle [66,67]. The fundamental geographic and climatic conditions in Newcastle, Australia, can be summarized as follows:

- The city is located at a latitude of 32.80 and a longitude of 151.83.
- The altitude is 33 m above sea level;
- The mean annual minimum and maximum temperatures are recorded as 14.3 °C and 21.8 °C, respectively;
- The yearly mean global radiation is measured at 4.8 kWh/m$^2$.

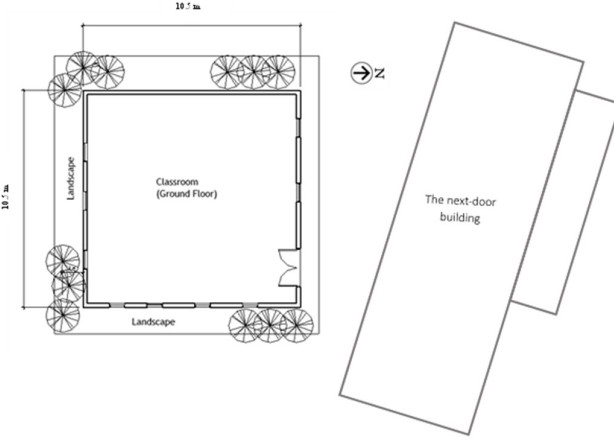

**Figure 2.** Ground-floor layout of the chosen educational facility (1:350 scale).

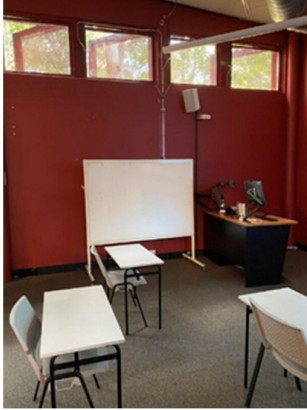

**Figure 3.** Classroom's interior design settings.

2.2.2. Field Investigation Data Collection

Many assumptions are typically made during modelling, mainly related to building services, system control, and operation. These assumptions may result in disparities between the forecasted and observed EC and TC [33]. Thus, it is essential to validate a simulation model before it is extensively adopted as a study tool. This section presents the validation approach, which was applied by comparing the simulation results with field investigation data, including student surveys, indoor physical measurements, and energy auditing.

A field study was undertaken to verify the results of the simulation for the baseline building model through the incorporation of both subjective and objective methods. The field investigation was an integral component of the present study's validation procedure, as detailed in [68]. The field investigations encompassed both subjective questionnaire surveys and objective physical measurements of indoor thermal comfort variables in three classrooms affiliated with The University of Newcastle. The adaptive thermal comfort (ATC) method was employed to calculate the actual mean vote (AMV) in accordance with ASHRAE Standard 55 for the subjective method [69]. Students were given the questionnaire during the last ten minutes of the lesson, simultaneously measuring the indoor thermal environment. The date and time of each response are recorded in a point-in-time survey, enabling comparisons with indoor environmental variables. The numerical scale is utilized to assess short-term indoor thermal comfort during the time of the lecture. The indoor thermal environment was measured objectively while participants simultaneously responded to questionnaires regarding their thermal comfort. During the surveyed classes, measurements were recorded for indoor air temperature ($T_a$), globe temperature ($T_g$), relative humidity ($RH$), and air velocity ($V_a$). The required minimum sample of students ($n$) was obtained using Sloven's Equation (1) [70]:

$$n = \left( \frac{N}{1 + Ne^2} \right) \tag{1}$$

where $N$ is the total number of students enrolled in the course who are assumed to use the classroom during lectures. The participants represent a sample with a 90% confidence level, which is conventional [71], and presents a 10% margin of error [72], denoted as $e$ in Equation (1). In total, 152 individuals participated in the field study, completing questionnaires while physical measurement of the indoor thermal environment were conducted.

*2.3. Building Performance and Evaluation*

The simulation process described in Section 2.2.1 was carried out for the reference building to simulate several ABDPs and assess the overall energy performance of the entire building. The ASHRAE heat balance method and the PMV-PPD method were selected as the calculation approaches in this study for evaluating EC and the level of thermal comfort, respectively. The PMV-PPD method, developed by Fanger (1970), serves as the basis for ISO 7730-2005 [73] and the ASHRAE 55-2010 standards [69]. PMV is a multi-condition variable index used to evaluate occupants' thermal comfort, employing a seven-point thermal sensation scale ranging from cold ($-3$) to hot ($+3$), with neutral (0) in the middle [74]. Environmental factors including $T_a$, $V_a$, mean radiant temperature ($MRT$), $RH$, as well as personal factors including the human metabolic rate and clothing insulation level, are considered as factors that can affect the level of thermal comfort [75]. To comply with ASHRAE 55 in mechanical ventilated buildings, the thermal limit on the seven-point scale of PMV is between $-0.5$ and $0.5$, regardless of the context of the building. TDHs in this study reflect the number of hours that the average PMV in the building is out of the occupants' thermal comfort range ($-0.5$ and $0.5$), implying a value of PPD higher than 20% [12]. Previous studies have used TDHs to denote the evaluation of thermal comfort [76–78]. The outcome variables used for this study were the EC (cooling/heating load) and student TDHs. These variables are considered important and widely used in measuring the energy performance and student thermal satisfaction in educational institutions [79,80].

### 2.4. Development of the ANN Model

An ANN is a forecasting method that allows the identification of complex non-linear relationships between a response variable and its predictors [81]. The ANN was chosen as its efficiency has been proven in many building studies [16,82–84]. Feedforward neural networks with back-propagation training were used in this study, as they have been commonly applied in engineering and environmental studies [25,85]. Using the gradient descent technique, back-propagation training seeks to limit the error function value in the weight space [86,87]. Therefore, the weights that minimise the error function are regarded as a learning problem solution [85]. Several software packages have been developed for ANNs. For MatLab, a widely used tool in research, a user-friendly and well-documented tool called neural network toolbox has recently been developed, which provides user codes with algorithms to set up, train, validate, and test ANNs [88]. Adequate training of an ANN system model requires an adequate representation of the available test data [89]. As demonstrated in Figure 4, Equation (2) relates every input and output [22]:

$$a = \text{logsig}\ (w_p + b). \tag{2}$$

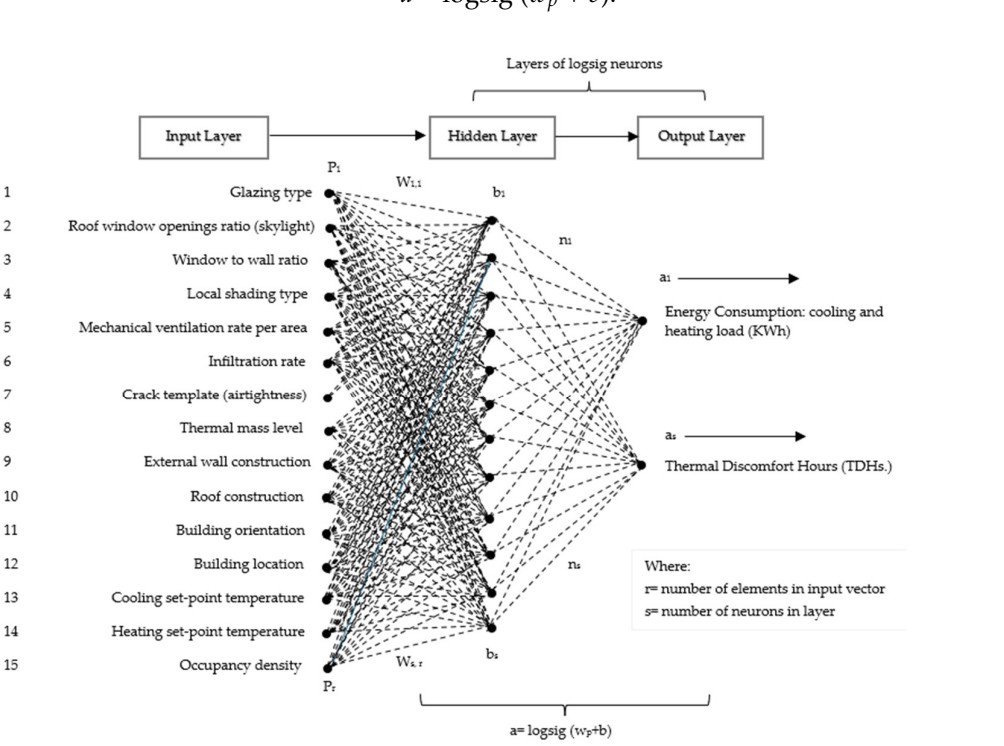

**Figure 4.** Proposed ANN model configuration.

In this linear equation, *a* is the output, *w* is the weight, *p* is the input, and *b* is the bias. At every stage, a set of these nodes is called a hidden layer. During the training of an ANN, the weight and bias of each node are determined using some converging algorithm, such that the output is brought as close to the actual output as possible. The accuracy of such models can be measured using metrics such as the root-mean-squared error between the predicted and actual outcomes.

Hence, building energy simulation (BES) was applied to generate broad scenarios of ABDPs and the associated EC and TDHs quantities (1846 scenarios). These scenarios were set up in the neural network toolbox, in order to develop an ANN model to predict EC and TC for any given combination of ABDPs. The proposed ANN model for this study is depicted in Figure 4. The specifics of the ANN are as follows:

- The parameters in Table 1 are represented by 15 input nodes, each of which represents a study parameter.
- The two output nodes represent each element of the objective function: EC and TDHs.

- There are 12 nodes in the hidden layer ($N_h$): there should be more than five hidden-layer neurons in an ANN [90]. The number of nodes required for this study was obtained using the following Equation (3) [91]:

$$N_h = 2/3 \ (\textit{inpuut layer size}) + (\textit{the size of the output layer}) = 2/3 \ (15) + (2) \tag{3}$$

where $N_h$ is the number of neurons in the hidden layer.

It is challenging to determine the number of cases needed to train an ANN model, as this depends on the statistical characteristics of the data and the complexity of the problem [92]. BES used LHS to extract 1846 scenarios of simulation data based on the model developed in this study. The training data in this study exceeded the minimum amount of training data, as computed with the following Equation (4) [93]:

$$N_d = \left( N_h - \frac{N_i + N_0}{2} \right)^2 \tag{4}$$

Here, $N_d$ represents the minimum required amount of training data, $N_h$ denotes the number of neurons in the hidden layer, $N_i$ indicates the number of the input layer, and $N_0$ stands for the number of neurons in the output layer. This significantly overestimated training data for the input cases, in agreement with previous studies [25,94,95]. To assess the performance of the ANN and explore whether its reliability is influenced by the sample size, this study adopted a method previously employed [90].

In terms of training and validating the ANN model to test the effects of the ABDPs, several studies of the machine learning literature have suggested approaches for data partitioning. The data set was divided as follows: a proportion of 70% for training, 15% for validation, and the remaining 15% of the total data set was allocated for testing [96–98]. The training data set comprises the data sample to train the model, whereas the validation data set is used to evaluate a given model and offer an objective evaluation of its fit on the training data [96,99]. The testing data set is used once a model is completely trained and validated, in order to provide an unbiased evaluation of a final model fit on the training data set [99,100]. The ANN model was also validated using the field investigation data.

For this study, 1291 scenarios were selected randomly [94] to train the ANN, 277 cases were used to validate the model, and the remaining 277 scenarios were used to test the model. An outcome variable (dependent variable) that included EC and TDHs was measured each time the independent variable was changed.

## 3. Results and Discussion

This section presents the results of the field investigation and validation of the simulation model. Data collected using the quantitative method are presented and interpreted in this section. The data indicate the student responses to the survey on the indoor thermal environment, thermal comfort evaluation, and indoor physical measurements. The actual measured data were then compared with simulation results. The results of the field investigation provide supporting evidence to validate the outcome of the simulation model.

In addition, this section discusses the development of the ANN model using the ANN Toolbox in MatLab. A MatLab application was then designed, which was used to fit the ANN model to a chosen data set. The application allows a user to select any parameter values as inputs and outputs, and can save ANN models which can be loaded later and used without having to train the model again. The application further allows a user to evaluate an ANN for a desired set of input parameters. This section presents the findings from the MatLab application using the developed ANN model to predict TDHs and total EC.

### 3.1. Validation of Indoor Thermal Environment

A field investigation was undertaken to validate the simulation outputs for the baseline building model through the incorporation of both subjective and objective methods, which was part of a previous study [68].

A comparison was made between the predicted value obtained from the simulation and the actual value of the field measurements, based on the subjective assessment of thermal comfort. The mean absolute error (MAE) and root-mean-square error (RMSE) metrics were employed for this comparison, in order to assess the precision of the simulation output and the degree of dispersion from the observed values, respectively. The MAE is calculated by averaging the absolute values of the discrepancies between the simulated and corresponding observations across the validation sample. RMSE is a quadratic scoring formula that assigns a comparatively high weight to substantial errors in order to quantify the average magnitude of error. Although there were certain distinctions in the indoor environmental variables, as reported in the field surveys and the BES outputs, the predictive mean vote (PMV) and actual mean vote (AMV) for student thermal comfort were comparable. Variations in the indoor environmental parameter simulation output and actual measurements were attributable to disparities in indoor heat gain factors, including equipment, illumination, and personnel. The existence of numerous air infiltration pathways within buildings facilitates the discharge of indoor thermal energy, potentially explaining the discrepancies observed between predicted and measured outcomes in classrooms [101]. Furthermore, according to some surveys, certain students lacked proper clothing for the classroom environment, regardless of whether it was cold or warm. There was a difference between the AMV and PMV, as the *clo* values differed between the simulation (which followed the international standard) and the field surveys. The field investigation (observed value) verified the BES model (predicted value), with error statistics indicating MAE = 0.156 and RMSE = 0.198. The MAE and RMSE values near zero indicate that the efficiency of the BES model had been enhanced, as detailed in Table 2.

**Table 2.** Comparison between the predicted values and observed values.

|  | $(O-E)$ [a] | $\lvert O-E \rvert$ | $(O-E)^2$ | MAE | MSE | RMSE |
|---|---|---|---|---|---|---|
| Total | −0.23 | 1.41 | 0.3561 | 0.156 | 0.039 | 0.198 |

Note: [a] O is the observed value, and E is the expected value.

After analysing the findings from the field investigation conducted in the classroom and comparing them with the output of the simulation model spanning a one-year period, the estimated percentage change from the current building to the simulation model was determined to be −13.5%. In the literature, a percentage difference (PD) of 15% or lower has been deemed acceptable when comparing the computer simulation results to actual data [102,103]. Consequently, the DB software yielded precise results in this investigation whenever the absolute PD value was 15% or lower. Numerous factors contribute to the disparity observed between the simulated and actual EC. Errors in the simulation may have resulted from disregarding air infiltration, climatic and environmental data, the pattern of window opening and closing, and the operation of heating and cooling systems [104,105].

### 3.2. Performance Analysis and Validation of the ANN with Multiple Output Variables

After the validation of the BES output, the ANN model was developed. It is essential to validate the developed ANN model's abilities to predict EC and TDHs through assessing the accuracy of the proposed ANN model. For this study, a data set containing over 2000 scenarios was obtained from the BES model using Monte Carlo analysis (MCA). A 70/15/15 split of the data set was used for training, validation, and testing [96–98]. A block diagram of the ANN is shown in Figure 5. The proposed ANN model consists of three layers: 15 input nodes, corresponding to the study parameters outlined in Table 1; 2 output nodes, representing EC and TDHs; and 12 nodes in the hidden layer.

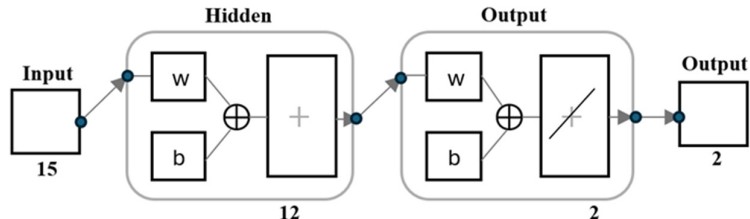

**Figure 5.** Block diagram for proposed ANN with 15 inputs, 2 outputs, and 12 hidden layers.

In this study, an ANN model comprising a feedforward neural network and a single hidden layer was designed. For back-propagation, the Levenberg–Marquardt algorithm was applied to train the ANN model [87]. Gradient descent was used to reduce the network error function, which has been successfully applied for ANN training in previous studies [90,106].

A training set of 1291 samples and a test set of 554 samples were used to create a statistical partition for the relative errors. Table 3 provides a summary of the relative error distributions for the two outcomes. It was observed that the total EC had a mean error of 1.37% and the TDHs output had a mean relative error of 0.22%. Average relative errors of less than 2% for total EC and less than 1% for average TDHs were achieved through using the ANN for simulation. However, reaching this accuracy required the use of 1291 training scenarios, corresponding to 270 times the number of input parameters. The average relative errors for EC and TDHs were considered to be good and acceptable, compared to other studies [16,90].

**Table 3.** Statistical repartition of relative errors in ANN testing.

| Relative Error | | <1% | <2.5% | <5% | <10% | <25% | Average (%) |
|---|---|---|---|---|---|---|---|
| Error rate of cases within the specified range | TDHs | 552 | 554 | 554 | 554 | 554 | 0.22 |
| | EC | 259 | 471 | 547 | 554 | 554 | 1.37 |

Network performance can be measured using the mean of squared errors (MSE) metric [107]. The regression plots demonstrate the effectiveness of training the model. An $R^2$ value close to one indicates that the utilised algorithms converged to a set of weights and biases for the neural network such that the error was minimized. In particular, the results were calculated using the Levenberg–Marquardt algorithm and different input–output configurations. The regression percentages for training and testing were in agreement with the $R^2$ for the output set to TDHs and EC, being higher than 0.97.

The comparison between the EC values predicted using the BES model (which was validated with the field investigation data) and the ANN model and the measured EC based on the field investigation of two classrooms at The University of Newcastle is presented in Table 4. According to the values of the statistical indicators, the discrepancy value for EC between the DB model and the field investigation for the two surveys was underestimated (−13.5% and −11.5%). On the other hand, the discrepancy value for EC between the ANN model and field investigation for the two surveys was overestimated (+9.5% and +7.8%). These findings show that the ANN model had higher accuracy across different ABDPs to predict EC in educational buildings in climate zone 5, when compared to the DB model.

**Table 4.** Measured and simulated energy consumption.

| | Field Investigation (kWh) | EnergyPlus Model | | ANN Model | |
|---|---|---|---|---|---|
| | | Simulation (kWh) | Discrepancy (%) | Simulation (kWh) | Discrepancy (%) |
| Survey 1 | 31,570 | 27,601 | −13.5 | 34,752 | +9.5 |
| Survey 2 | 36,806 | 32,813 | −11.5 | 39,829 | +7.8 |

*3.3. Implementation of ANN Model*

In order to enable individuals without prior knowledge of modelling and simulation tools to utilise the ANN model for prediction of EC and TDHs, a user-friendly interface was developed. Subsequently, the proposed model was utilized in a case study depicting an educational building situated in climate zone 5 in Australia. Numerous combinations of building design alternatives and scenarios can be assessed with the developed model. For scenarios in which a significant amount of computational time is needed to process many alternative combinations, collaboration between modelling and simulation tools is even more important. The design team encountered challenges when attempting to balance ABDPs in order to generate a group of design options, which are subsequently simulated in order to determine EC and TDHs. Utilising the developed interface, predictions were generated for EC and TDHs across 15 distinct combinations of building design options. These options are characterised by variations in baseline dimensions and additional input parameters such as the roof type, local shading type, thermal mass level, and temperature set-point.

The proposed interface is expected to contribute to the implementation of the developed ANN model and facilitate its operation as a decision support tool. This interface is straightforward, reliable, and immediately applicable, requiring no prior knowledge of modelling and simulation tools. The interface has multiple text boxes, panels, and buttons, which allow the user to train and test ANN models. The data file name is a text field which shows the path of the data file (Microsoft Excel file) loaded by the user. Every time the 'Load Data' button is pressed, this field is updated. The input box contains all the input variables present in the Excel file. The block is initially empty upon startup. As soon as the user selects a data file using the load data button, the inputs box is loaded with the input variables. The user can choose multiple inputs by holding the 'Ctrl' button and selecting the desired inputs. The output textbox file contains the output variable names. As with the inputs, the user has the option of selecting multiple output variables. This button is disabled by default. Once the data have been loaded, the button becomes active and can be used to train an ANN. For training, the app uses a function called *trainANN.m*, which takes the input and output data vectors/arrays and returns a trained model and its performance characteristics. Hence, complex non-linear relationships between parameters can be identified through the use of this model, which is frequently challenging to accomplish with conventional regression techniques.

**4. Conclusions**

The thermal performance and design of educational buildings have significant impacts on energy consumption and thermal comfort. Key decisions influencing sustainability in building performance are often made during the conceptual design stage, involving a comparison of various architectural building design parameters and comprehensive thermal property data. Utilising a combination of computer-based simulation tools enables designers and engineers to evaluate potential decisions and achieve long-term goals more effectively. These simulations offer a numerical approach for the prediction of building performance, circumventing the need for costly and time-consuming full-scale field experiments. To aid decision-makers and designers in developing effective designs, a standardised model for evaluating energy performance is essential. This model is expected to accurately forecast both thermal comfort and energy consumption in educational buildings.

Our study employed a building energy simulation analysis using DB software, Monte Carlo analysis to encompass all probable input scenarios for the selected architectural building design parameters (ABDPs), and artificial neural networks (ANNs) to assess occupant thermal discomfort hours (TDHs) and energy consumption (EC). Validation was achieved through comparing the simulation outputs with real field data. For a warm temperate climate in New South Wales, Australia, we developed, trained, and tested an intelligent computational model. The $R^2$ value exceeded 0.97, indicating the model's ability to minimise the error through determining the optimal weights and biases for the neural network. The regression percentages for training and testing aligned well with the $R^2$ values set for TDHs and EC. The mean errors for total EC and TDHs were found to be 1.37% and 0.22%, respectively, validating the model's accuracy and suitability for a MatLab-based design application. When comparing the discrepancies between the simulation models (DB and ANN) and field data in three classrooms, the intelligent computational model exhibited better prediction accuracy for ABDPs in energy consumption, with +9.5% and +7.5% overestimations, compared to the DB model's underestimations of −13.5% and −11.5%. This indicates that the proposed model could minimise the prediction error by about 50% for educational buildings in climate zone 5.

## 5. Limitations and Further Study

This study specifically presents an intelligent computational model, addressing 15 combinations of ABDPs and their distributions to refine EC and thermal comfort predictions. The distribution characteristics of the ABDPs were derived from the possible values in architectural practice revealed through the literature review. In this study, only a single-storey education building in climate zone 5 with a mechanically ventilated mode was considered, whereas the results from considering other building types, ventilation modes, and climate zones might be different.

Further studies should focus on the subjects suggested by the limitations of the present study. It is recommended to use the same study methodology, changing the input parameters by introducing more ABDPs and climate zones to enhance the thermal building performance. The next stage of the research may involve more active and passive design strategies, including developing a database to include different climate zones for the extensive use of thermal performance predictions. It is also suggested to focus on personal and environmental thermal comfort factors by offering a personal climate control solution for both air-conditioned and free-run buildings to enhance indoor environmental quality. The EN15251 standard specifies the indoor environmental input parameters for designing and assessing the energy performance of buildings, covering indoor air quality, thermal conditions, lighting, and acoustics [108]. Therefore, an assessment of the building's long-term efficiency concerning these four indoor environmental input parameters is needed. The applications of ANN and other intelligent computational algorithms need further exploration for their use in building performance predictions.

**Author Contributions:** Conceptualisation, S.A., W.T., S.K. and D.A.; methodology, S.A., W.T., S.K. and D.A.; software, S.A. and D.A.; validation, S.A., W.T., S.K. and D.A.; formal analysis, S.A.; investigation, S.A.; resources, S.A., W.T., S.K. and D.A.; data curation, S.A.; writing—original draft preparation, S.A. and W.T.; writing—review and editing, S.A., W.T., S.K. and D.A.; visualisation, S.A. and W.T.; supervision, W.T., S.K. and D.A.; project administration, S.A. All authors have read and agreed to the published version of the manuscript.

**Funding:** This research received no external funding.

**Institutional Review Board Statement:** Not applicable.

**Informed Consent Statement:** Not applicable.

**Data Availability Statement:** Dataset available on request from the authors.

**Acknowledgments:** The first author gratefully acknowledges the Saudi Arabian Cultural Mission (SACM) in Australia and the scholarship sponsored by Imam Abdulrahman Bin Faisal University (IAU) for financially supporting this research.

**Conflicts of Interest:** The authors declare no conflicts of interest.

## Abbreviations

| | |
|---|---|
| ABDPs | Architectural building design parameters |
| AMV | Actual mean votes |
| ANN | Artificial neural network |
| ASHRAE | The American Society of Heating, Refrigerating, and Air Conditioning Engineers |
| ATC | Adaptive thermal comfort |
| BES | Building energy simulation |
| Clo | Clothing insulation (1 Clo = 0.155 m$^2$·K/W) |
| DB | DesignBuilder |
| EC | Energy consumption (kWh) |
| EPW | EnergyPlus Weather |
| hrs | Hours |
| HVAC | Heating, ventilation and air conditioning |
| LHS | Latin hypercube sampling |
| MAE | Mean absolute error |
| MCA | Monte Carlo analysis |
| MRT | Mean radiant temperature (°C) |
| MSE | Mean squared error |
| PD | Percentage difference (%) |
| PMV | Predicted mean vote |
| PPD | Predicted percentage dissatisfaction |
| $R^2$ value | Coefficient of determination |
| RH | Relative humidity (%) |
| RMSE | Root mean square error |
| $T_a$ | Air temperature (°C) |
| TDHs | Thermal discomfort hours |
| $T_g$ | Globe temperature (°C) |
| $T_o$ | Operative temperature (°C) |
| UA | Uncertainty analysis |
| $V_a$ | Air velocity (m/s) |

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
