# Peer review of "Optimising Building Energy and Comfort Predictions with Intelligent Computational Model"

_sustainability, doi:10.3390/su16083432_

Round 1

Reviewer 1 Report

Comments and Suggestions for Authors

General comment

The paper proposes an ANN for optimising energy and comfort predictions in buildings. The paper is well structured, but there are some issues that should be addressed.   Introduction - "Implementing effective energy consumption methods in sectors with high demand" - what is the energy consumption method? - The paragraph on ANN is quite long. I recommend reducing it. - Currently, the paper lacks a section on data-driven thermal comfort modelling, that should be taken into account. For instance, you may refer to https://doi.org/10.1016/j.enbuild.2020.110392 and 10.1016/j.totert.2023.100083   Materials and methods:  - In general, please avoid being too wordy, try to be more concise in the definition of the methodology - In the paper, there are several acronyms, which make quite difficult the reading of the text. I suggest reducing the acronyms to enhance comprehension of the paper and adding a list of symbols at the beginning of the paper. - How were the ABDPs selected? How were the ranges/values of these parameters selected? - Which is the U-value considered for external walls, roof, and glazing types? - Section 2.2 looks more like an introduction section than a methodological one. Please explain it more clearly. - Please specify what is the "e" in Eq. 1. Moreover, which subjective measurements were assessed? - Section 2.3 should be better explained (methods with which the building simulation was carried out)   Results and discussion - Please divide these two sections. In the results, just report the information, while in the discussion analyse and compare them also with previous research - Paragraph 3.1 should be summarized  - Limitations of the study should be added   Conclusions: future directions of research should be included

Reviewer 2 Report

Comments and Suggestions for Authors

This paper presents an intelligent computational model, addressing various combinations of architectural building design parameters and their distributions. It will promote the field of energy efficiency in buildings. To further improve the article, the following are suggestions to consider:

1Page 6, Figure 2 and Page 9, Figure 4 are not clear and it is recommended that the author replace them with vector graphics.

2There are inconsistencies in the spacing of formulas and context in the article, and the author is advised to check and revise.

3Are "input layer size" and "the size of the output layer" in equation (3) on page 8, line 301 inappropriate? It is suggested that the authors replace them with variables.

4Some positions in Table2 on page 11 do not have values, and it is suggested that the authors make additions.

5The thermal comfort predictions considered in the article are clearly related to the outside temperature, but do not take into account the outside uncertainty. It is suggested that the authors can refer to DOI: 10.17775/CSEEJPES.2021.04510, which adds scenario-based stochastic planning to the model. It is hoped that the authors can add this challenge to future research.

Comments on the Quality of English Language

Minor editing of English language required

Reviewer 3 Report

Comments and Suggestions for Authors

MDPI Sustainability Journal (Manuscript ID: sustainability-2925070)

Comments to the Author

This paper presents a computational model using Monte Carlo, EnergyPlus and ANN to refine energy consumption and thermal comfort predictions. The authors are presenting a useful research topic and the paper is well-written. However, there are several points that need to be addressed to improve the quality of the manuscript.

Suggestions to improve the quality of the paper are provided below:

1.     There is an apparent concern within the manuscript regarding similarities with the authors' previous work. Notably, sections such as the introduction (lines 107-113) and the validation of the indoor thermal environment (section 3.1) exhibit resemblances, with a high similarity score of 31%, which is much higher than acceptable. However, upon closer examination, I find no evidence of outright unoriginality or duplication in the submitted work. I believe this manuscript explores a distinct research application, methodology, and results compared to the authors' previous work. Therefore, rather than opting for immediate rejection, I would like to provide the authors with an opportunity to restructure the overlapping parts carefully before resubmission.

2.     In line 47, the authors mention energy savings can be achieved while improving thermal comfort predictions. To further elaborate on this aspect, I suggest integrating more recent literature and practical machine-learning applications of thermal comfort models in this area. One pertinent topic is definitely 'personal thermal comfort models' [ https://doi.org/10.1016/j.buildenv.2018.01.023]. These models offer the capability to predict each occupants’ thermal comfort preferences, thereby contributing to both energy efficiency and overall comfort levels. Especially, the recent advancements in practical machine learning applications have yielded notable developments in data efficiency of the model predictions [DOI: https://doi.org/10.1016/j.buildenv.2023.110148] and energy efficiency in HVAC systems [DOI: https://iopscience.iop.org/article/10.1088/1742-6596/2600/13/132004/meta]. Please mention these to enrich the introduction of machine learning & thermal comfort background (or future works) and provide readers with a comprehensive and up-to-date understanding of the area.

3.     In the conclusion, please mention the limitations of the proposed methodology and how it can be improved as part of future work. Also, mention the future directions.

Round 2

Reviewer 2 Report

Comments and Suggestions for Authors

This paper presents an intelligent computational model, addressing various combinations of architectural building design parameters and their distributions. It will promote the field of energy efficiency in buildings. To further improve the article, the following are suggestions to consider:

1On page 3, line 72, the [22] font size does not match the other character font sizes, and the author is asked to harmonise.

2Page 11, Table 2 has a vacant position, please make additions.

3The article mentions the next phase of research focusing on personal thermal comfort factors, available at DOI: 10.17775/CSEEJPES.2021.04510, which analyses in detail the impact of comfort temperature on building energy management.

Comments on the Quality of English Language

Minor editing of English language required

Reviewer 3 Report

Comments and Suggestions for Authors

Thank you for addressing my comments and suggestions. This manuscript is ready for publication.

Author Response

Thank you for reviewing our paper and confirming that it is ready for submission.